# Crosstalk between β2- and α2-Adrenergic Receptors in the Regulation of B16F10 Melanoma Cell Proliferation

**DOI:** 10.3390/ijms23094634

**Published:** 2022-04-22

**Authors:** Paola Matarrese, Sonia Maccari, Barbara Ascione, Rosa Vona, Vanessa Vezzi, Tonino Stati, Maria Cristina Grò, Giuseppe Marano, Caterina Ambrosio, Paola Molinari

**Affiliations:** 1Center for Gender-Specific Medicine, National Institute of Health, Viale Regina Elena 299, 00161 Rome, Italy; paola.matarrese@iss.it (P.M.); sonia.maccari@iss.it (S.M.); barbara.ascione@iss.it (B.A.); rosa.vona@iss.it (R.V.); tonino.stati@iss.it (T.S.); 2National Center for Drug Research and Evaluation, National Institute of Health, 00161 Rome, Italy; vanessa.vezzi@iss.it (V.V.); mariacristina.gro@iss.it (M.C.G.); caterina.ambrosio@iss.it (C.A.); paola.molinari@iss.it (P.M.)

**Keywords:** cyclic AMP, β2-adrenergic receptors, α2-adrenergic receptors, proliferation, melanoma

## Abstract

Adrenergic receptors (AR) belong to the G protein-coupled receptor superfamily and regulate migration and proliferation in various cell types. The objective of this study was to evaluate whether β-AR stimulation affects the antiproliferative action of α2-AR agonists on B16F10 cells and, if so, to determine the relative contribution of β-AR subtypes. Using pharmacological approaches, evaluation of Ki-67 expression by flow cytometry and luciferase-based cAMP assay, we found that treatment with isoproterenol, a β-AR agonist, increased cAMP levels in B16F10 melanoma cells without affecting cell proliferation. Propranolol inhibited the cAMP response to isoproterenol. In addition, stimulation of α2-ARs with agonists such as clonidine, a well-known antihypertensive drug, decreased cancer cell proliferation. This effect on cell proliferation was suppressed by treatment with isoproterenol. In turn, the suppressive effects of isoproterenol were abolished by the treatment with either ICI 118,551, a β2-AR antagonist, or propranolol, suggesting that isoproterenol effects are mainly mediated by the β2-AR stimulation. We conclude that the crosstalk between the β2-AR and α2-AR signaling pathways regulates the proliferative activity of B16F10 cells and may therefore represent a therapeutic target for melanoma therapy.

## 1. Introduction

Melanoma develops in melanocytes, which are the cells that produce melanin, and is the most aggressive and life-threatening skin cancer. The limited success of classic chemotherapeutics has led to the introduction of targeted therapies aimed at tackling detected oncogenic mutations and/or other DNA aberrations underlying melanomagenesis. Several novel drugs that target the MAPK pathway, the kinase MEK, a member of the MAPK family, or immune checkpoints, key regulators of the immune system, have been developed [1]. However, melanoma resistance or recurrence after targeted therapies advocates for further research to unveil novel regulators of melanoma growth and additional effective therapeutics in melanoma patients.

In recent years, preclinical and clinical studies have provided evidence that catecholamines, which are released in response to physical or emotional stress, can affect the growth and metastasis of many types of cancer, including melanoma. Once released, catecholamines, such as norepinephrine and epinephrine, can activate both β- and α-adrenergic receptors, promoting the physiological responses that follow stimulation of the sympathetic nervous system. β-adrenergic receptors (β-AR), which belong to the guanine nucleotide-binding G protein-coupled receptor superfamily, are expressed in human tissues, such as benign melanocytic naevi, atypical naevi and malignant melanoma, as well as in B16F10 melanoma, the most frequently used syngeneic murine melanoma, as evidenced by qPCR, Western blot and cytofluorimetric analyses [2,3,4,5], and have emerged as important mediators of catecholamine effects on melanoma growth. For example, treatment with propranolol, a β1- and β2-AR antagonist, attenuates cancer growth, both in preclinical models of melanoma and in patients with melanoma, suggesting that β-AR stimulation may regulate melanoma growth [4,6,7,8,9,10,11,12]. Additionally, in a B16F10 melanoma model, combined treatment with epinephrine and propranolol decreases the in vitro proliferative activity of cancer cells, suggesting a potential modulating role of the β-AR signaling pathway in the antiproliferative action of the catecholamine epinephrine [2]. However, given that catecholamines can stimulate all nine AR subtypes and, in the presence of propranolol, can activate not only subtypes of α-ARs but also β3-ARs, which may produce effects opposite to those of β1 and β2-AR stimulation [13], the relative role of each β-AR subtypes in regulating cancer cell proliferation, as well as their contribution to the antiproliferative action of α-ARs on B16F10 cells, remain to be clarified.

In a recent study [2], we reported that B16F10 melanoma cells express both α2A- and α2B-AR subtypes and that the treatment with ST-91, an α2B-AR agonist, reduces the in vitro proliferation, survival and mitochondrial function of B16F10 cultured cells. The α2-AR agonists have been used for decades to treat arterial hypertension and, in more recent years, as adjuncts for sedation and to reduce anesthetic requirements. However, whether α2-AR agonists exert a class effect in inhibiting B16F10 melanoma cell proliferation remains to be ascertained.

The objective of this study was threefold: first, determining whether tumor β-ARs are functioning, that is, whether their stimulation causes an increase in cAMP levels; second, determining whether clonidine, the prototypical α2-AR agonist acting on all three α2-AR subtypes, affects B16F10 cell proliferation; third, assessing whether β-AR stimulation by the catecholamine isoproterenol modulates the antiproliferative action of clonidine on B16F10 cells and, if so, evaluating to what extent each β-AR subtype contributes to regulating the proliferative activity of B16F10 cells. Using Ki-67 protein staining, luciferase-based cAMP assay, flow cytometry and B16F10 melanoma cells, we found that activation of β2-ARs with isoproterenol increases cAMP levels in cancer cells, indicating that these receptors are functioning. We also found that β-AR subtype stimulation, on its own, has no effect on cancer cell proliferation, suggesting that β-AR activation has no direct tumor-promoting effects. Finally, we observed that the stimulation of the β2-AR suppresses the antiproliferative effects of α2-AR stimulation on B16F10 cells, whereas both the β1-AR and β3-AR appear to play only a marginal role. Thus, the present study establishes a crosstalk between the β2-AR and α2-AR signaling pathways that is involved in regulating B16F10 melanoma cell proliferation.

## 2. Results

### 2.1. cAMP Accumulation in B16F10 Cells

B16F10 cells express β1-, β2- and β3-ARs, as evidenced by qPCR, Western blot and cytofluorimetric analyses [2,3,4,5]. Here, we used the live-cell biosensor GloSensor (Promega, Madison, WI, USA) to assess whether tumor β-ARs are coupled to the Gs-cAMP signaling pathway, that is, whether their stimulation causes an increase in cAMP levels. The cells were treated with isoproterenol (a β1-, β2- and β3-AR agonist), propranolol (a β1-β2 antagonist), ICI 118,551 (a β2 selective antagonist), or isoproterenol plus either propranolol or ICI 118,551. Furthermore, cAMP levels were evaluated after stimulation of α2-AR with the agonists clonidine and ST-91.

The cAMP levels increased after isoproterenol treatment (Figure 1A,B). Agonist stimulation of the β-ARs activates cellular signaling, primarily through the G protein-mediated generation of intracellular second messengers, such as cyclic adenosine monophosphate (cAMP), by the effector enzyme adenylyl cyclase. However, within a few minutes, the cAMP levels essentially returned to the unstimulated state despite the continued presence of the drug. The transient cAMP response to isoproterenol is not unique to B16F10 cells. A similar transient cAMP response to isoproterenol was reported in HEK 293 cells [14] and in rat vascular smooth muscle cells, indicating that cAMP dynamics are also rapid in other cells. Propranolol and ICI 118,551 treatments completely suppressed the isoproterenol effects on cAMP levels, suggesting that these are mediated by stimulation of β1- and/or β2-ARs, and that β3-ARs negligibly contribute to changes in the cAMP levels in B16F10 cells (Figure 1B). We also performed new experiments to identify the β-AR subtype involved in the regulation of B16F10 cell proliferation, comparing the rank order of potency of three catecholamines (isoproterenol, norepinephrine and epinephrine) on cAMP production. The rank order of affinity or potency is often used as a “fingerprint” to differentiate receptors. The rank order of affinity for the β2-AR is isoproterenol > epinephrine >> norepinephrine, whereas for the β3-AR, it is isoproterenol ≥ norepinephrine >> epinephrine. By constructing concentration–response curves for isoproterenol, epinephrine and norepinephrine (Figure 1C), we found that the rank order of the potency of catecholamines for cAMP enhancement was isoproterenol > epinephrine >> norepinephrine, which is the typical signature of the β2-AR [15,16]. Since the β1-AR is not present in the cells used in this study [11], these results are compatible with the notion that the β2-AR is the predominant β-AR in B16F10 cells (Figure 1C). Finally, the α2 agonists clonidine and ST-91 do not affect cAMP levels, because the α2-ARs are Gi-coupled receptors and inhibit adenylyl cyclase activity and cAMP production.

### 2.2. Effect of α2-AR Agonists on B16F10 Cell Proliferation

In a previous paper, we reported the antitumor effect of α2-AR stimulation [5]. In particular, we demonstrated that treatment with ST-91, an α2-AR agonist that activates the Gi protein pathway, significantly reduced the proliferation of cultured murine B16F10 melanoma cells. To ascertain whether the effects of ST-91 on tumor proliferation are shared by other α2-agonists, we also evaluated the effects of clonidine (CLO), the prototype of α2-AR agonists. The proliferative activity of the B12F10 melanoma cells, as assessed by Ki-67 staining, significantly decreased after treatment for 48 hours with both ST-91 (about −29%, *p* < 0.05 vs. CTR) and clonidine (CLO, about −24%, *p* < 0.05 vs. CTR) compared to control (Figure 2A–C). Moreover, although the α2-AR antagonist yohimbine (YOH) did not induce per se any effect on cell proliferation (about +3% YOH vs. CTR, *p* > 0.05), it prevented both ST-91 (ST-91 + YOH about −30% vs. ST-91, *p* < 0.05) and clonidine (CLO + YOH about −25% vs. CLO, *p* < 0.05) antiproliferative effects (Figure 2A–C).

Since we could not rule out an effect of ST-91 and CLO also on α1-AR, at least at the concentration of 1 μM, we also used YM-254890 (YM), a selective inhibitor of the Gq protein subfamily. In fact, the α1-AR is associated with the Gq protein, unlike the α2-AR, associated with the Gi protein, and this allowed us to discriminate the effects related to the activation of the two receptors. We did not observe any effect on proliferation, either in cells treated with YM or when YM was administered in dual treatment with ST-91 or CLO (Figure 2A, YM + ST-91 vs. ST-91, and YM + CLO vs. CLO, *p* > 0.05). This would seem to indicate that the effect of these drugs was essentially mediated through the α2-AR.

Collectively, these results suggest that the α2-AR agonist class is able to modulate the proliferative activity of cultured B16F10 melanoma cells (Figure 2).

### 2.3. Effect of β-AR Agonists on α2-Agonists-Induced Cell Proliferation Inhibition

To assess the relative role of each β-AR subtype in modulating the antiproliferative action of the α2-AR agonists, B16F10 cells were treated for 48 h with the following AR ligands: isoproterenol (ISO), ST-91, clonidine (CLO), propranolol (PRO), ICI 118,551 (ICI), isoproterenol plus propranolol (PRO), clonidine plus isoproterenol, or ST-91 plus isoproterenol (Figure 3). We found that isoproterenol alone had no effects on B16F10 cell proliferation, as assessed by Ki-67 staining (MFI 27.58 ± 3.27), compared to control (MFI 29.7 ± 3.45, *p* > 0.05).

Given that β3-ARs may produce effects opposite to those of β1- and β2-AR stimulation [13], we tested the effects of a combined treatment with isoproterenol and propranolol, a β1- and β2-AR antagonist. No effect on cell proliferation was observed (MFI 27.98 ± 2.72, *p* > 0.05 vs. CTR) compared to control, suggesting that β3-AR stimulation does not regulate cell proliferation on its own under these experimental conditions (Figure 3A).

Next, we evaluated the effects of β-AR stimulation on the antiproliferative properties of α2-AR stimulation (Figure 3B). We found that isoproterenol at the concentration of 1 µM suppresses the antiproliferative effects of both ST-91 (MFI 30.1 ± 2.1, *p* < 0.05 ST-91 + ISO vs. ST-91) and clonidine (MFI 27.4 ± 4.5, *p* < 0.05 CLO + ISO vs. CLO) on B16F10 cells (Figure 3B). In turn, the suppressive effects of isoproterenol were abolished by the treatment with either ICI 118,551 (MFI 27.4 ± 4.5), a β2-AR antagonist, or propranolol, suggesting that isoproterenol effects are mainly mediated by β2-AR stimulation (Figure 3C).

### 2.4. Effect of cAMP Levels on B16F10 Cell Proliferation

Finally, we evaluated whether the suppressive effect of isoproterenol is attributable to an increase in cAMP levels. To this aim, the cells were treated with forskolin, a direct adenylyl cyclase activator, or IBMX, a non-selective phosphodiesterase inhibitor. We paradoxically found that both forskolin and IBMX significantly inhibit cell proliferation on their own by about 40% (Figure 4).

Since recent advances in cAMP reporter technology have enabled direct evidence of cAMP compartmentalization, our results suggest that isoproterenol may regulate B16F10 cell proliferation by acting on a compartmentalized β2-AR-cAMP-dependent signaling pathway rather than on the global pool of cAMP.

## 3. Discussion

In recent years, β-ARs, which are expressed in human melanoma cells as well as in murine B16F10 melanoma cells, have emerged as important mediators of catecholamine effects on melanoma growth. This suggests the possibility that β-AR blockers, a class of drugs primarily used for antagonizing catecholamine effects on the cardiovascular system, may provide new therapeutic strategies for the control of melanoma progression.

In the present study, we show that the stimulation of tumor β2-ARs causes an increase in cyclic AMP levels without modifying the proliferative activity of B16F10 cells, as evidenced by the analysis of the Ki-67 protein. This implies that catecholamine stimulation of the β2-AR-Gs-cAMP signaling pathway has no pro-tumorigenic activity on its own in this cellular model of melanoma. An equally important result of this study is that β2-AR stimulation, although not showing tumor-promoting activity, suppresses the antiproliferative activity associated with α2-AR stimulation, highlighting a crosstalk between the two adrenergic signaling pathways. Indeed, isoproterenol stimulation inhibited the antiproliferative effects of both ST-91 and clonidine, two different α2-AR agonists. We also demonstrate that this inhibitory effect is mediated by the β2-AR stimulation with a molecular mechanism that remains to be ascertained. Finally, we show that the increase in global intracellular cAMP levels through the stimulation of adenylyl cyclase with forskolin or the nonselective inhibition of phosphodiesterases with IBMX causes a reduction in the proliferative activity of B16F10 melanoma cells. These results are not surprising. Indeed, to facilitate the specificity of cAMP-dependent signaling pathways, these are compartmentalized by A-kinase anchoring proteins and phosphodiesterases [17,18]. Thus, different cAMP pools have different effects on B16F10 melanoma cell proliferation, and enhancing some cAMP pools rather than others may have therapeutic implications for melanoma treatment.

Murine B16F10 melanoma cells express both β2- and β3-Ars, as demonstrated by qPCR, Western blot and cytofluorimetric analyses [2,3,4,5]. Both β2- and β3-ARs are coupled with Gs proteins, the activation of which increases cAMP levels, and both these receptors are involved in melanoma progression. Indeed, treatment with β-blockers, such as propranolol, a β-blocker targeting β1- and β2-ARs, or SR59230A and L-748337, both β3-AR antagonists, appears effective in reducing tumor growth in a mouse model of melanoma, as well as in increasing the overall survival of melanoma patients [2,3,4,5,6,7,8,11,12,19,20]. In vitro studies have shown that treatment with propranolol, β3-AR antagonist L-748337 or β3-AR siRNAs reduces proliferation and induces apoptosis of human and mouse melanoma cells [5,20,21], whereas β3-AR stimulation with BRL37344 promotes melanoma cell proliferation and reduces apoptosis [22]. However, despite the huge amount of research, it is unclear whether this involvement is attributable to cAMP-dependent signaling pathways. In the present study, we show for the first time that β2-AR stimulation causes an increase in cAMP levels, whereas β3-AR stimulation does not increase it. Since β3-AR stimulation may produce effects opposite to those of β1- and β2-AR stimulation through Gi-protein coupling [13], it is possible that β3-ARs in B16F10 cells are coupled with Gi-protein, which would explain the lack of cAMP increase after treatment with isoproterenol. This is in agreement with previous results showing that β3-AR stimulation produces nitric oxide-mediated effects [22]. Regarding the effects of β3-AR stimulation with β-AR agonists on in vitro proliferation, we found that, contrary to what was previously reported using BRL37344, isoproterenol does not affect B16F10 cell proliferation. This discrepancy between these two β3-AR agonists could be attributable to the different pharmacological properties of these β-AR ligands. In this perspective, it is known that BRL37344, but not isoproterenol, does not cause β-AR desensitization [23]. These findings require further investigation.

The present study also extends the current knowledge of crosstalk between β- and α2-ARs by demonstrating that β2-AR stimulation negatively regulates the effects mediated by the stimulation of α2-AR subtypes on B16F10 cell proliferation. Crosstalk involving these receptors has been known for a long time. Since β- and α2-ARs couple to G proteins with opposing actions on adenylyl cyclase activity, stimulation of the α2-ARs can antagonize β2-AR signaling. In addition, it has been shown that α2-AR stimulation with clonidine enhances the sensitivity of β-AR signaling in astrocytes [24] and that the pharmacological properties of α2-ARs are regulated by β-ARs [25,26,27].

It is worth commenting on an important message that emerges from this study. Isoproterenol suppresses the effects of α2-AR stimulation on cancer cell proliferation while both forskolin, a direct activator of adenylyl cyclase, and IBMX, a non-selective phosphodiesterase inhibitor, albeit with different mechanisms of action, inhibit cancer cell proliferation on their own. In this regard, it should be noted that the second cAMP messenger is synthesized in response to a series of extracellular stimuli that produce very different biological effects. Recent studies have provided an insight into how cAMP-dependent signals retain their specificity, demonstrating that the spatial and temporal dynamics of cAMP are regulated by A-kinase anchoring proteins and discreetly positioned phosphodiesterases that act as sinks. Therefore, our results suggest that isoproterenol can regulate the proliferation of B16F10 cells by acting on a compartmentalized β2-AR-cAMP-dependent signaling pathway, and that the enhancement of some cAMP pools rather than others may inhibit the growth of B16F10 melanoma.

In conclusion, the results of this study reveal a crosstalk between β2-AR and α2-AR that may play a key role in the control of B16F10 melanoma cell proliferation.

## 4. Materials and Methods

### 4.1. Cell Cultures

A B16F10 cell line was cultured at 37 °C in humidified atmosphere containing 5% CO_2_. The low-passage B16F10 murine melanoma cell line, obtained from ATCC, was maintained in DMEM high-glucose medium (EuroClone, West York, UK) supplemented with 5% FBS (EuroClone, West York, UK) in the presence of penicillin and streptomycin (Sigma-Aldrich, St. Louis, MO, USA).

### 4.2. cAMP Measurements Using GloSensor cAMP Assay

For cAMP experiments, B16F10 cells were infected with Puro-resistant retrovirus encoding the luciferase-based probe GloSensor 22 F, followed by puromycin selection (5 μg/mL). The luciferase-based intracellular cAMP probe GloSensor 22 F was purchased from Promega (Madison, WI, USA) and subcloned into puromycin resistance retroviral expression vector pQCXIP (Clontech, Mountain View, CA, USA).

To select cell lines with optimal expression of the biosensor, virally transduced cells were plated at very low density in 15 cm dishes, and 50–100 individual clones from each cell line were isolated with silicon rings for further screening. The cell clones with the best response to forskolin were identified by luminescence recording in 96-well plates as described [28].

B16F10 cells stably expressing GloSensor-22F probe were seeded into 96-well white plastic plates at a density of 2 × 104 cells/well and grown for an additional 24 h. For cAMP assay, the wells were washed once with PBS and further incubated for 60 min in 50 μL PBS containing 25 mM glucose and 2 mM luciferin. Next, 50 μL PBS containing 10 μM rolipram without or with β-adrenergic ligands (isoproterenol 1 μM, propranolol 10 μM, isoproterenol plus propranolol) diluted in 0.1% BSA in PBS or 100 μM forskolin, used as a positive control, were added to the wells, and the plates were transferred in a luminometer (Victor Light, PerkinElmer, Milan, Italy). Luminescence from each well was counted every 30 s with 0.5 s integration time for about 80 min. cAMP response was determined as area under the luminescence–time curves (Figure 1A).

### 4.3. Cell Treatments

Mouse melanoma B16F10 cells were cultured as described above. For proliferation experiments, the cells were treated for 48 h with 1 μM of the β-AR agonist isoproterenol (ISO), the β1- and β2-AR antagonist propranolol, the β2-AR antagonist ICI 118,551 (ICI), the α2-AR agonists ST-91 (2-[2,6-diethylphenylamino]-2-imidazoline) and clonidine, or the α2-AR antagonist yohimbine (YOH), alone or in different combinations. In another set of experiments, the cells were treated with the non-selective phosphodiesterase inhibitor IBMX (1 μM) or the adenylyl cyclase activator forskolin (FSK, 1 μM) for 48 h. To exclude the involvement of the α1-AR in the antiproliferative effects induced by ST-91 and clonidine, we added YM-254890 (1 μM), a selective inhibitor of the Gq protein subfamily able, therefore, to selectively inhibit α1-AR (associated with Gq), to ST-91 or CLO, respectively. Cells left untreated or treated with a drug’s vehicle were considered as the control (CTR). Adrenergic ligands were purchased from Sigma-Aldrich (St. Louis, MO, USA), except for ST-91 and YM-254890 (R&D Systems, Minneapolis, MN, USA).

### 4.4. Proliferation Assay

Proliferative activity was analyzed by Ki-67 nuclear antigen expression using the fluorescein isothiocyanate (FITC) mouse anti-human Ki-67 after cell fixation and permeabilization according to the manufacturer’s protocol (BD Biosciences, San Jose, CA, USA). All samples were acquired on a FACSCalibur flow cytometer (BD Biosciences, San Jose, CA, USA) equipped with a 488 argon laser and a 635 red diode laser, and at least 10,000 events per sample were run. Data were analyzed using the Cell Quest Pro software (BD Biosciences, San Jose, CA, USA). Results were reported as median fluorescence intensity (MFI).

### 4.5. Immunofluorescence Analysis

For nuclear detection of Ki-67, control and treated cells were fixed in acetone/methanol 1/1 (*v*/*v*) for 10 min at room temperature and air-dried. After 1 h of preincubation with PBS containing 10% of AB human serum, the cells were incubated for 1 h at room temperature with FITC-conjugated mouse anti-human Ki-67 (BD Biosciences). The nuclei were stained with Hoechst 33258 (Sigma-Aldrich) at 37 °C for 15 min. The samples were mounted on glass cover slips with glycerol/PBS (2:1) and observed by intensified video microscopy (IVM) with an Olympus Microphot fluorescence microscope (Olympus Corporation, Tokyo, Japan) equipped with a Zeiss CCD camera.

### 4.6. Data and Statistical Analysis

Data are expressed as mean ± SD and were analyzed using GraphPad Prism version 5.03 software (GraphPad Software Inc., San Diego, CA, USA). Unpaired t test was performed to accept or reject the null hypothesis. Differences with *p* < 0.05 were considered significant.

## Figures and Tables

**Figure 1 ijms-23-04634-f001:**
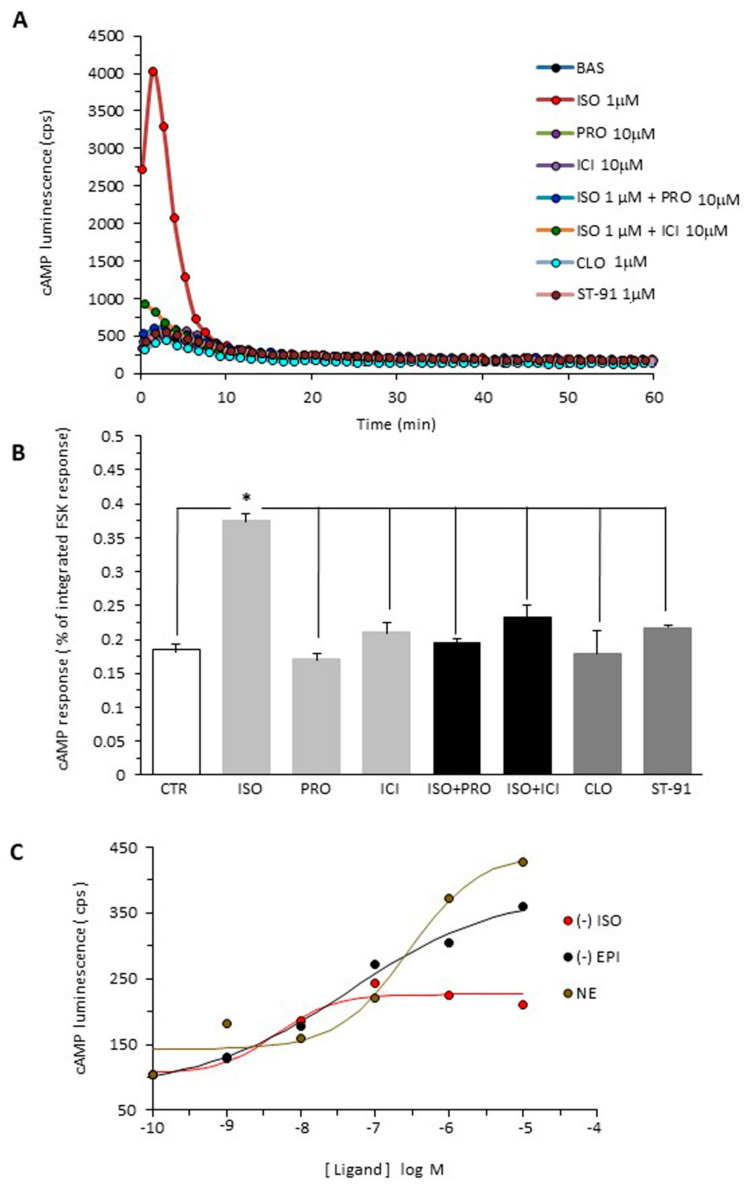
Effects of β-AR and a-AR ligands on cAMP production in B16F10 cells: (**A**) Representative image of cAMP measurements in real time using a GloSensor cAMP biosensor (bas, baseline). (**B**) Integrated cAMP responses computed as % of integrated forskolin response from tracings obtained in B16F10 cells stably expressing GloSensor-22F probe. Measurements were obtained in the absence or the presence of the β-AR agonist isoproterenol (ISO 1 μM), the antagonists propranolol and ICI 118,551 (PRO 10 μM, ICI 118,551 10 μM), isoproterenol plus either propranolol or ICI 118,551 (ISO 1 μM + PRO 10 μM or ICI118,551 10 μM), and the α2 agonists clonidine (CLO 1 μM) and ST91 (ST-91 1 μM). Data of three independent experiments are reported. * *p* < 0.05. (**C**) Concentration–response curves for the ligand-induced enhancement of cAMP production. Epinephrine (EPI), isoproterenol (ISO), norepinephrine (NE). EC_50_ for ISO, EPI and NE was 4.6 nM, 45 nM and 284 nM, respectively.

**Figure 2 ijms-23-04634-f002:**
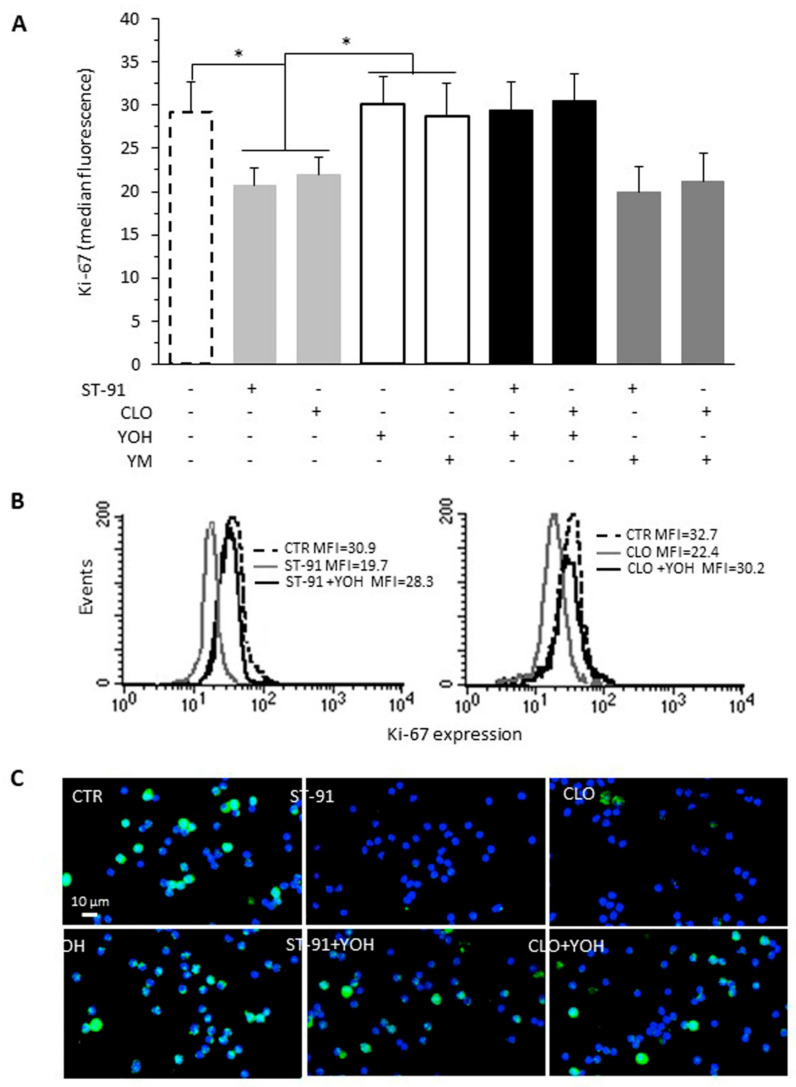
Effect of α2-AR agonists on B16F10 cell proliferation. Cell proliferation was evaluated by flow cytometry measuring Ki-67 nuclear antigen expression in B16F10 cells, untreated (CTR) or treated with clonidine (CLO, 1 μM) or ST-91 (1 μM) alone or in combination with the α2-AR antagonist yohimbine (YOH, 1 μM) or with a Gq protein inhibitor YM-254890 (YM) for 48 h (left panel). (**A**) Bar graph showing the results obtained from five different experiments and expressed as mean ± SD. On the ordinate y-axis, the median fluorescence intensity values are reported. * *p* < 0.05. (**B**) Cytofluorimetric curves showing one representative experiment out of five. Numbers represent the median fluorescence intensity. (**C**) Representative micrographs of B16F10 cells, treated as indicated, after dual staining with an FITC-conjugated antibody against Ki-67 (green) and with Hoechst (blue).

**Figure 3 ijms-23-04634-f003:**
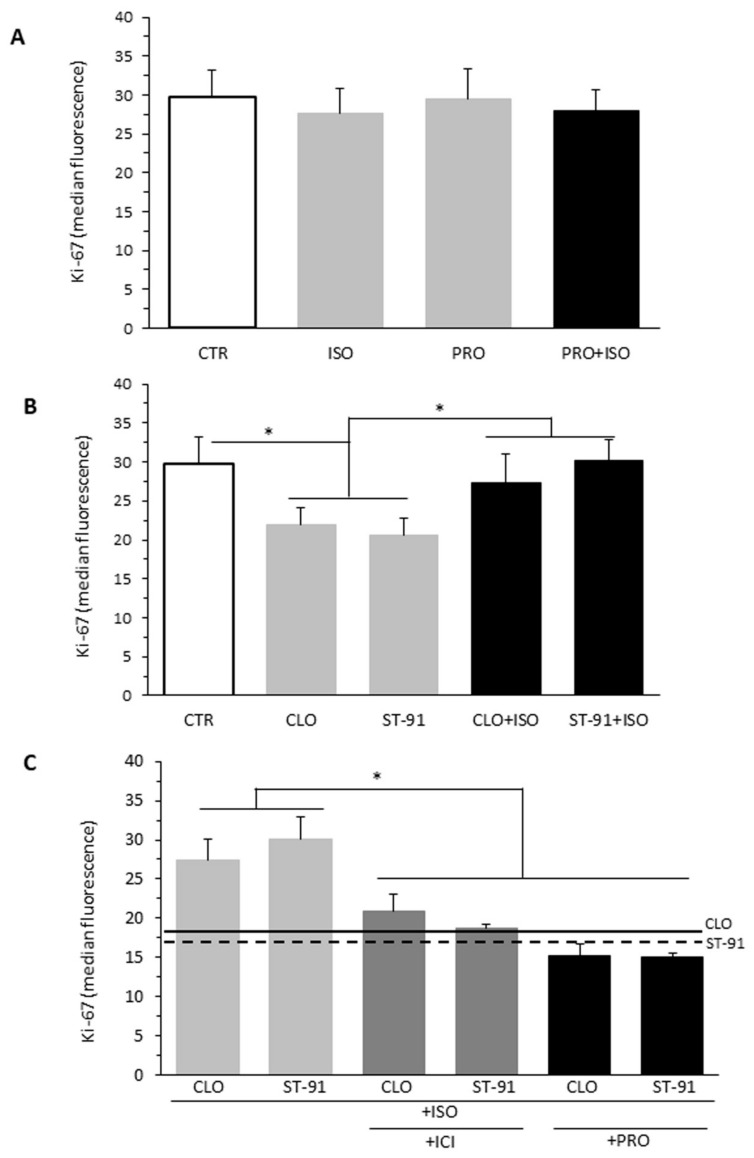
Effect of β-AR agonists on α2-agonists-induced cell proliferation inhibition. Flow cytometry evaluation of cell proliferation performed by measuring Ki-67 nuclear antigen expression in B16F10 cells in different experimental conditions. (**A**) Cells untreated (CTR) or treated with isoproterenol (ISO, 1 μM), propranolol (PRO, 10 μM), or ISO plus PRO for 48 h. (**B**) Cells untreated (CTR) or treated with clonidine (CLO, 1 μM) or ST-91 (1 μM) alone or in combination with isoproterenol (ISO, 1 μM). (**C**) Cells treated with CLO (1 μM) plus ISO (1 μM) or ST-91 (1 μM) plus ISO (1 μM) in the presence or the absence of ICI (1 μM), a selective β2-AR antagonist, or PRO (1 μM), a non-selective β-blocker. Data from four independent experiments are reported. On the ordinate y-axis, the median fluorescence intensity values are reported. Solid and dashed lines indicate the proliferation values in cells treated with CLO or ST-91 alone, respectively. * *p* < 0.05.

**Figure 4 ijms-23-04634-f004:**
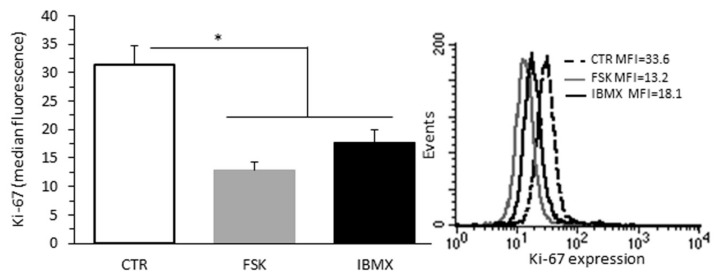
Effect of cAMP levels on B16F10 cell proliferation. (**Left** panel) Flow cytometry analysis of cell proliferation after staining with Ki-67 nuclear antigen of B16F10 cells untreated or treated with FSK, a direct adenylyl cyclase activator, or IBMX, a non-selective phosphodiesterase inhibitor. Data from three independent experiments are reported. (**Right** panel) On the ordinate y-axis, the median fluorescence intensity values are reported. Results from one representative experiment out of three are shown, * *p* < 0.05.

## Data Availability

All data generated or analyzed during this study are included in the present article.

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
