# Peer review of "Crosstalk between β2- and α2-Adrenergic Receptors in the Regulation of B16F10 Melanoma Cell Proliferation"

_ijms, 2022, doi:10.3390/ijms23094634_

Round 1

Reviewer 1 Report

Review comments

The research article by Paola Matarrese on “Crosstalk between β2- and α2-adrenergic receptors in the regulation of B16F10 melanoma cell proliferation” is an interesting subject. Some of my Major concerns are:

  1. In Figure, the authors have suggested that isoproterenol treatment increased the levels of cAMP, which lasted for 10 minutes, and after that I cannot find any significance difference from baseline, ISO, PRO or ISO+PRO groups after 10 minutes. Please explain.
  2. What were the dose of ISO and PRO used to acquire the results of Figure 3A and 3B? In Fig. 3A, we see that there is no difference between ISO or PRO groups in comparison to control on the concentration of Ki67 marker. However, in Figure 3B, we observed that co-treatment with CLO+ISO and ST-91+ISO significantly upregulated the expression of ki67. I do not understand this situation. Why wouldn’t ISO treated groups have more Ki67 expression than control groups in Fig 3A. please explain the rationale behind it.
  3. In Line number 168-169 the authors have mentioned that Isoproterenol may regulate B16F10 cell proliferation acting on a compartmentalized β2-AR-cAMP-dependent signaling pathway. The authors need to provide more necessary data to prove this claim.
  4. Why were the levels of cAMP not evaluated for ST-91, clonidine and ICI 118,551.
  5. The authors need to provide a graphical abstract for this study as well.
  6. At this point the data provided is not enough and sufficient to warrant a publication to the journal. I request the authors to perform more studies.
  7. The authors should also provide morphological images of B16F10 cells after treatment with the different drugs.
  8. Immunofluorescence staining images for the expression of Ki67 and cAMP after treatment with the different drugs used for the study would be a great addition to this study.

Minor comments

  1. I request the authors to make changes in Figure 1A. and make the different groupings as Line-plots.
  2. I request the authors to provide good high-quality images as the image quality is really bad.

Author Response

Reviewer 1

Review comments

The research article by Paola Matarrese on “Crosstalk between β2- and α2-adrenergic receptors in the regulation of B16F10 melanoma cell proliferation” is an interesting subject.

A.: We would like to thank the reviewer for the positive feedback.

Major concerns

Reviewer

  1. In Figure, the authors have suggested that isoproterenol treatment increased the levels of cAMP, which lasted for 10 minutes, and after that I cannot find any significance difference from baseline, ISO, PRO or ISO+PRO groups after 10 minutes. Please explain.

A.: Agonist stimulation of βARs activates cellular signaling primarily through the G protein-mediated generation of intracellular second messengers such as cyclic adenosine monophosphate (cAMP), by the effector enzyme adenylyl cyclase. However within a few minutes the cAMP levels return essentially to the unstimulated state despite the continued presence of the drug. The transient cAMP response to isoproterenol is not unique to B16F10 cells. A similar transient cAMP response to isoproterenol was reported in HEK 293 cells [14], and in rat vascular smooth muscle cells, indicating that cAMP dynamics are also rapid in other cells. This information has been added to the revised manuscript (line 98-106)

Reviewer

  1. What were the dose of ISO and PRO used to acquire the results of Figure 3A and 3B? In Fig. 3A, we see that there is no difference between ISO or PRO groups in comparison to control on the concentration of Ki67 marker. However, in Figure 3B, we observed that co-treatment with CLO+ISO and ST-91+ISO significantly upregulated the expression of ki67. I do not understand this situation. Why wouldn’t ISO treated groups have more Ki67 expression than control groups in Fig 3A. Please explain the rationale behind it.

 A.: As for the concentrations, we used the 1 μM concentration both for ISO and PRO, as reported in figure 3 legend (Panel A). We found that ISO or PRO per se did not induce any alteration in the proliferation of B16F10 melanoma cells. However, when administered in co-treatment with ST-91 or clonidine (CLO), the β-agonist ISO was able to prevent the antiproliferative effects of both these drugs (Panel B). The administration of the β2-antagonist, ICI 118551 or that of the non-selective β-AR  antagonist PRO were able to inhibit the activity of ISO against ST-91 and clonidine, thus restoring the antiproliferative activity of clonidine and ST-91. This would seem to indicate that the stimulation of the β2-AR, while inducing an increase in cAMP levels, did not modify the proliferation of B16F10 melanoma cells. However, although β2-AR stimulation did not show per se protumoral activity, it suppressed the antiproliferative activity associated with α2-AR stimulation.

Reviewer

  1. In Line number 168-169 the authors have mentioned that Isoproterenol may regulate B16F10 cell proliferation acting on a compartmentalized β2-AR-cAMP-dependent signaling pathway. The authors need to provide more necessary data to prove this claim.

 A.: We thank the reviewer for this comment. In the present study, we found that the increase in global intracellular cAMP levels through the stimulation of adenylyl cyclase with forskolin or the nonselective inhibition of phosphodiesterases with IBMX causes a reduction in proliferative activity of B16F10 melanoma cells. On the contrary, we also found that the increase of cAMP levels by isoproterenol has no effect on the growth of B16F10 cells. A possible but not the only explanation for these discrepant effects is that, to facilitate the specificity of cAMP-dependent signaling pathways, these are compartmentalized by A-kinase anchoring proteins and phosphodiesterases [14,15]. Thus, different cAMP pools have different effects on B16F10 melanoma cell proliferation and enhancing some cAMP pools rather than others may have therapeutic implications for melanoma treatment. Part of this discussion has been added to the revised version (please see Discussion section, line 297-305).

Reviewer

  1. Why were the levels of cAMP not evaluated for ST-91, clonidine and ICI 118,551.

 A.: As required, we have carried out additional experiments to evaluate the effects of ST-91, clonidine and ICI 118,551 on cAMP production. The data were included in the new Figure 1.

Reviewer

  1. The authors need to provide a graphical abstract for this study as well.

 A.: As required, we added the graphic abstract of the study, which schematically shows the data obtained.

Reviewer

  1. At this point the data provided is not enough and sufficient to warrant a publication to the journal. I request the authors to perform more studies.

 A.: We thank the reviewer for this suggestion. Following the indications of the reviewers, we have carried out some additional experiments to better demonstrate our hypothesis, namely that β2-AR stimulation suppresses α2-AR-associated antiproliferative activity in the B16F10 murine melanoma cell line.

In particular,  we added:

(i) analysis of cell proliferation in the presence of the Gq protein inhibitor YM-254890 (YM), alone or in combination with ST-91 or CLO. Given that the α1-AR is associated with the Gq protein whereas the α2-AR is associated with the Gi protein, this experiment has allowed us to dissect the contribution of the two α-ARs. We observed no effect on proliferation neither in cells treated with YM nor when YM was administered in dual treatment with ST-91 or CLO (new Figure 2A, YM+ST-91 vs. ST-91, and YM+CLO vs. CLO, p>0.05). This would seem to indicate that the effect of ST-91 and CLO was essentially mediated through the α2-AR. (please see the revised version, line 165-172).

(ii) immunofluorescence images of B16F10 cells, untreated or treated with different drugs after labeling with Ki-67 (new Figure 2C).

(iii) additional experiments to evaluate the effects of ST-91, clonidine and ICI 118,551 on cAMP   production (figures 1A-B)

(iv) concentration–response curves for catecholamine-induced enhancement of cAMP production (figure 1C)

Reviewer

  1. The authors should also provide morphological images of B16F10 cells after treatment with the different drugs. Immunofluorescence staining images for the expression of Ki67 and cAMP after treatment with the different drugs used for the study would be a great addition to this study.

 A.: As required, immunofluorescence images of B16F10 cells, untreated or treated with different drugs after labeling with Ki-67, were included in the new Figure 2.

Minor comments

Reviewer

  1. I request the authors to make changes in Figure 1A and make the different groupings as Line-plots.

 A.: We changed the figure 1A as requested.

Reviewer

  1. I request the authors to provide good high-quality images as the image quality is really bad.

 A.: Thanks for this comment. We think that the poor quality of the images may depend on the conversion system of the word file to PDF, as the images we inserted in the template are 600dpi images. This problem can be easily overcome by submitting the figures as single files.

We would like to thank the reviewer again for taking the time to review our manuscript.

Reviewer 2 Report

This study demonstrates a link between the sympathetic nervous system and melanoma. The design of the study  is straightforward but the data presented does not support the conclusion, since the choice of ligands and especially their concentrations used here makes a clearcut interpretation impossible.  

  1. Studies have shown that alpha 1 adrenergic receptor (PMID: 15278367) agonists decrease cell proliferation in melanoma. In this study authors investigate the effects of alpha 2 agonists on cell proliferation. However, clonidine and ST-91 also show significant binding at alpha 1 AR (PMID: 34355529). The same is true for the antagonist yohimbine (PMID: 32608144). In the concentrations used in this study, clonidine, ST-91 and yohimbine are all capable to bind and activate alpha 1 adrenoreceptors! Therefore, a dose-response curve needs be established and alpha 1 contribution should be excluded!
  2. The authors described propranolol as specific for β1/2. That is not true. Propranolol has moderate affinity for β3 (PMID: 15655528). The KD value is around 100nM for β3, which means, that at 10µM it blocks pretty much all beta adrenergic effects. Therefore, a dose-response curve needs be established!
  3. The same is true for ICI 118551; at 1µM, all beta ARs will be inhibited by this compound (PMID: 15655528). To specifically block beta 2, [c] should be around 10nM! Therefore, a dose-response curve needs be established!

In general, choice of ligands should be re-evaluated

Author Response

Reviewer 2

Review comments

This study demonstrates a link between the sympathetic nervous system and melanoma. The design of the study is straightforward but the data presented does not support the conclusion, since the choice of ligands and especially their concentrations used here makes a clearcut interpretation impossible.  

We thank the reviewer for pointing out the potential value of our work and for her/his insightful comments and efforts towards improving our manuscript. We amended it to add new experimental data. We hope to have adequately addressed the reviewer’s comments and concerns and we present our reply to each of them separately (the reviewer’s comments are in italics).

Reviewer

  1. Studies have shown that alpha 1 adrenergic receptor (PMID: 15278367) agonists decrease cell proliferation in melanoma. In this study authors investigate the effects of alpha 2 agonists on cell proliferation. However, clonidine and ST-91 also show significant binding at alpha 1 AR (PMID: 34355529). The same is true for the antagonist yohimbine (PMID: 32608144). In the concentrations used in this study, clonidine, ST-91 and yohimbine are all capable to bind and activate alpha 1 adrenoreceptors! Therefore, a dose-response curve needs be established and alpha 1 contribution should be excluded!

 A.: We agree with the issue raised by this reviewer. To answer this question, we performed additional experiments using YM-254890, a selective inhibitor of the Gq protein subfamily. In fact, the α1AR is associated with the Gq protein whereas the α2AR is associated with the Gi protein. This allowed us to dissect the contribution of the two α-ARs. The results obtained from these experiments have highlighted a negligible effect of α1ARs in the inhibition of proliferation induced by treatment with ST-91 and clonidine on B16F10 melanoma cells, which would therefore seem essentially attributable to α2AR stimulation. This information has been added to the revised version (line 165-172).

Reviewer

  1. The authors described propranolol as specific for β1/2. That is not true. Propranolol has moderate affinity for β3 (PMID: 15655528). The KD value is around 100nM for β3, which means, that at 10µM it blocks pretty much all beta adrenergic effects. Therefore, a dose-response curve needs be established!

 A.: We agree with the reviewer. The concentrations used do not allow to separate the contribution of the different β-ARs. However, in our opinion, the problem can be overcome by identifying the predominant β-AR subtype by well-known pharmacological approaches. Therefore, we have carried out new experiments to identify the β-AR subtype involved in the regulation of B16F10 cell proliferation comparing the rank order of potency of three catecholamines (isoproterenol, norepinephrine, and epinephrine) on cAMP production. The rank order of affinity or potency is often used as a “finger print” to differentiate receptors. The rank order of affinity for β2-AR is isoproterenol>epinephrine>>norepinephrine, whereas for the β3-AR it is isoproterenol≥norepinephrine>>epinephrine. By constructing concentration-response curves for these catecholamines (figure 1C), we found that the rank order of the potency of catecholamines for cAMP enhancement was isoproterenol > epinephrine >> norepinephrine, which is the typical signature of the β2-AR (Hoffmann et al., Naunyn Schmiedeberg's Arch Pharmacol, 2004, 369, 151-9; Baker JG, Br J Pharmacol 2010, 160, 1048-1061, doi: 10.1111/j.1476-5381.2010.00754.x). Since the β1AR is not present in the cells used in this study (Maccari et al, BJP 2017), these results are compatible with the notion that the β2AR is the predominant βAR in B16F10 cells (Fig 1C). This information has been added to the revised version (line 109-121).

Reviewer

  1. The same is true for ICI 118551; at 1µM, all beta ARs will be inhibited by this compound (PMID: 15655528). To specifically block beta 2, [c] should be around 10nM! Therefore, a dose-response curve needs be established!

 A.: As above mentioned, the results of concentration-response curves for catecholamines are compatible with the notion that the β2-AR is the predominant βAR in B16F10 cells (Fig 1C).

Reviewer

  1. In general, choice of ligands should be re-evaluated

 A.: We thank the reviewer for this comment. In the new version of the paper, we have better defined the adrenergic ligands that we used in our experiments. To dissect the different contribution given by the α1- and α2-ARs to the antiproliferative effect observed in the B16F10 murine melanoma cells, we have also included the results obtained in the presence of YM-254890, a selective inhibitor of the Gq protein subfamily. Please see the revised version, line 165-172.

We would like to thank the reviewer again for taking the time to review our manuscript.

Round 2

Reviewer 1 Report

The authors have answered all the questions that I have raised during my first review. I have no other questions. Accept it for publication.